# Successful Intraosseous (IO) Adenosine Administration for the Termination of Supraventricular Tachycardia (SVT) in a 3.5-Year-Old Child—Case Report and Literature Review

**DOI:** 10.3390/healthcare12151509

**Published:** 2024-07-30

**Authors:** Jakub Zachaj, Łukasz Kręglicki, Tomasz Sikora, Katarzyna Moorthi, Filip Jaśkiewicz, Klaudiusz Nadolny, Robert Gałązkowski

**Affiliations:** 1Medical Emergency Department, Medical University of Warsaw, Litewska 14/16, 00-575 Warsaw, Poland; robert.galazkowski@wum.edu.pl; 2Polish Medical Air Rescue, Księżycowa 5, 01-934 Warsaw, Poland; l.kreglicki@lpr.com.pl (Ł.K.); t.sikora@lpr.com.pl (T.S.); k.moorthi@lpr.com.pl (K.M.); 3Department of Emergency Medicine, Jagiellonian University Medical College, Michałowskiego 12, 31-126 Cracow, Poland; 4Department of Emergency Medicine, Faculty of Medical Sciences in Katowice, Medical University of Silesia, Poniatowskiego 15, 40-055 Katowice, Poland; 5Military Medical Institute, National Research Institute, Szaserów 128, 04-141 Warsaw, Poland; 6Emergency Medicine and Disaster Medicine Department, Medical University of Lodz, Pomorska 125, 90-419 Lodz, Poland; filip.jaskiewicz@umed.lodz.pl; 7Faculty of Medicine, Department of Emergency Medical Service, Silesian Academy in Katowice, ul. Rolna 43, 40-555 Katowice, Poland; klaudiusz.nadolny@akademiaslaska.pl

**Keywords:** supraventricular tachycardia, intraosseous, adenosine, emergency, paediatrics

## Abstract

Paediatric supraventricular tachycardia (SVT) is a common arrhythmia of great clinical significance. If not treated promptly, it can cause heart failure and cardiogenic shock. Depending on the patient’s condition, SVT treatment involves vagal manoeuvres, pharmacological, or direct current cardioversion. The goal of acute SVT management is to immediately convert SVT to a normal sinus rhythm (NSR) and prevent its recurrence. Adenosine is recommended as the first-line treatment for stable SVT by the European Resuscitation Council (ERC) and American Heart Association (AHA) guidelines, when vagal manoeuvres have proven ineffective. The ERC and AHA guidelines recommend the intravenous route of administration. The intraosseous (IO) administration technique is also possible, but still relatively unknown. The aim of this paper is to describe a 3.5-year-old child with SVT that was converted to NSR following IO administration of adenosine. Successful conversion was achieved after the second attempt with the adenosine dose. In the described case, there was no recurrence of SVT.

## 1. Introduction

Supraventricular tachycardia (SVT) is the most common cardiac arrythmia among paediatric populations, affecting 1 in 250 to 1000 children [1]. SVT is defined as a rapid and regular heart rate, often exceeding 180 beats per minute (bpm) in children, and 220 bpm in infants. SVT originates above the His bundle [2]. SVT is a broad group of tachyarrhythmias that includes atrioventricular (AV) node-dependant tachyarrhythmias, such as accessory pathway-mediated atrioventricular reciprocating tachycardia (AVRT), as well as atrial tachycardias such as atrial fibrillation, atrial flutter, and ectopic atrial tachycardia [3]. In the case of supraventricular arrhythmias of focal origin, the cause is pathological (increased) automaticity of a specific group of cardiomyocytes or the so-called triggered activity, i.e., early or late depolarization of cardiomyocytes. An alternative mechanism of arrhythmia development is the macro-re-entry mechanism, i.e., the circulation of a depolarization wave stimulating previously depolarized areas that have already regained excitability. Focal arrhythmias caused by increased automaticity or triggered activity are also referred to as non-re-entrant. Focal arrhythmias also include arrhythmias occurring in the micro-re-entry mechanism (short re-entry loop) [4].

The management of SVT depends on the patient’s condition [5]. For patients in stable clinical condition with SVT, treatment includes vagal manoeuvres, followed by adenosine if the vagal manoeuvres fail to convert SVT to NSR. Patients in unstable clinical condition with SVT are treated with electrical cardioversion. Adenosine is not effective in the treatment of atrial flutter, atrial fibrillation, and ventricular tachycardia; however, there are rare types of ventricular tachycardias that are adenosine-sensitive (idiopathic VT).

In infants, the signs and symptoms of SVT may be nonspecific and include symptoms such as poor feeding, vomiting, diaphoresis, hypersomnia, and irritability. Toddlers and school-aged children may present with classic cardiac symptoms such as palpations, chest pain, shortness of breath, and syncope. Adolescents may present with this constellation of symptoms, accompanied by anxiety and decreased exercise tolerance. Before initiating appropriate SVT management, factors such as exertion, anxiety, stress, pain, respiratory failure, fever, infections, sepsis, hypoglycaemia, poisoning, hypovolemia, pulmonary embolism, anaphylaxis, anaemia, etc., must be ruled out [6].

## 2. Case Report

The Helicopter Emergency Services (HEMS) was dispatched to a 3.5-year-old child with arrhythmia. The ground Emergency Medical Team (EMS) on site requested support for patient management and securing their transport to a specialist centre. Once they arrived at the HEMS landing site by ambulance, the patient was lying on a stretcher, was verbally responsive (AVPU), drowsy, and breathing on their own at a rate of approximately 30 breaths per minute on passive oxygen therapy (straight mask) with a flow of 5 litres per minute. The pulse, palpated at the carotid artery, was approximately 200 beats per minute. No monitoring or IV access was established. In a repeated assessment, the patient remained verbally responsive (AVPU—V), the airways were patent and not at risk; there were no secretions in the nasal cavity or oral cavity; respiration was efficient at 24 breaths per minute; there was no shortness of breath or respiratory effort and no central cyanosis; there was slight cyanosis on the periphery (phalanges); SpO_2_ was 81–85% (HEMS increased oxygen flow to 10 L/min); auscultatory vesicular murmur was symmetrical bilaterally. SVT with approximately 235 bpm was diagnosed (the ECG recording, on the basis of which SVT was diagnosed, was sent in paper form along with the patient to the hospital, it was not possible to restore the recording from the ground ambulance device’s memory); NBP was 144/88; capillary refill was normal; GCS was 12 (E-4, V-4, M-5); glycemia was normal; pupils were equal and reactive to light; there were no meningeal symptoms, convulsions, visible injuries or skin lesions; and the temperature was normal. According to the information provided by the mother, the child had cardiological problems (the child’s guardian was unable to determine what diseases the child was suffering from because he was under the influence of alcohol and strong emotional disturbances) and was treated at a specialist clinic. For about 40 min, the child presented with a very rapid heart rate, was crying at first, and was then apathetic. During the ABCDE examination, two attempts to establish peripheral vascular access were unsuccessful. Due to the distance from the specialist centre and air transport conditions (limited space and no possibility of performing some procedures during the flight), and after consultation with the specialist centre, a decision was made to try to stabilize the patient on site. For this purpose, IO access was established by means of the EZ-IO system [Teleflex Medical Triangle Park, NC] with a 15 mm needle inserted into the right tibia near the tibial tuberosity. Once IO access was confirmed, 5 mL of 0.9% NaCl was administered first, followed by lidocaine 2% at a dose of 0.5 mg/kg body weight (8 mg IO). As this failed to alleviate the patient’s pain, a second dose of lidocaine 2% was administered, constituting 1/2 of the first dose (4 mg IO). Adenosine was then prepared at a dose of 0.1 mg/kg body weight (1.5 mg), which was administered at 12:59 (the event was recorded in the attached heart monitor tracing). The technique used to deliver adenosine at the scene was the double-syringe technique [DST] (Figure 1). The rhythm temporarily went from 233 bpm to 203 bpm. At 13:05, a decision was made to administer the second dose of 0.2 mg/kg body weight (3 mg) of adenosine. At the next monitoring interval (2 min), the heart rate was 143 bpm; after another 10 min, the patient’s heart rate normalized (to 128 bpm) (Figure 2). In both administrations, the adenosine was not diluted. For faster administration, a fluid bolus of 5 mL of 0.9% NaCl was then pushed. After reassessing the patient’s condition, the patient was transferred from the ambulance to the helicopter. During transport, the patient was stable, with no recurrence of SVT. The patient was transferred in a good general condition to the staff at the Hospital Emergency Department (ED).

## 3. Discussion

As mentioned previously, the choice of SVT management depends on the patient’s hemodynamic and clinical condition (patients in stable clinical condition with SVT vs. patients in unstable clinical condition with SVT). Clinical findings in SVT may differ, depending on the age of the child and the duration of SVT. ERC and AHA guidelines recommend vagal manoeuvres and adenosine as the first-line treatment for stable SVT [7,8].

Adenosine can serve as a diagnostic or therapeutic agent. In diagnostics, adenosine is used in myocardial perfusion stress imaging by virtue of its vasodilatory effects. In a therapeutic setting, adenosine is used for its antiarrhythmic properties in supraventricular tachycardia and can function as a diagnostic tool, depending on the type of SVT, in both paediatric and adult patients [9]. Adenosine can also unmask (though not terminate) atrial flutter in cases of unclear tachycardia, if used for diagnostic issues. Adenosine is indicated as an adjunct to thallium-201 in myocardial perfusion scintigraphy, as well as for the conversion of paroxysmal supraventricular tachycardia into sinus rhythm. Adenosine has a short duration of action (half-life <10 s) and a wide therapeutic window [10]. Agonism of adenosine receptors A1 and A2 reduces conduction time in the atrioventricular node of the heart. Conduction velocity is decreased by inducing potassium efflux and inhibiting calcium influx through channels in the nerve cells, leading to hyperpolarization and an increased threshold for calcium-dependent action potentials. The decreased conduction time leads to an antiarrhythmic effect. Adenosine has an effect on the cells of the AV node. Due to its effect on the AV node, Adenosine can only terminate re-entrant SVT involving the AV node in the re-entry circuit (AVRT and AVNRT) but not in cases of intra-atrial re-entry or focal atrial tachycardia. The inhibition of calcium influx reduces the activity of adenylate cyclase, relaxing the vascular smooth muscles. Relaxed vascular smooth muscles lead to increased blood flow through normal coronary arteries but not through stenotic arteries, allowing for the easier uptake of thallium-201 in normal coronary arteries [10,11]. The most frequently described side effects of adenosine administration are bronchoconstriction, seizures, and hypersensitivity. Adenosine can also cause bradycardia, leading to *torsade de pointes*, especially in patients with a prolonged QT interval. A total of six cases of arrhythmia caused by the administration of adenosine in paediatric patients (aged 0–16 years with overt or hidden Wolf–Parkinson–White syndrome [WPW]) have been described in the literature. The cases mentioned included three cases of atrial fibrillation, two cases of atrial flutter and one case of ventricular fibrillation. In three cases, the arrhythmias resolved spontaneously, while in the remaining cases they required treatment with amiodarone or amiodarone administration and electrical cardioversion [12]. Patients with an overdose of adenosine may present with asystole, heart block, or cardiac ischemia. These effects are generally short-lived. Patients with an overdose should receive symptomatic and supportive care, which may include a slow intravenous injection of theophylline [9,10].

Adenosine failure occurs when its administration does not result in a definitive or sustained termination of tachycardia. There are three scenarios that cause this, as follows: administration errors (e.g., insufficient adenosine dose), failure to terminate the arrhythmia, and insufficient duration of the adenosine effect [13].

Adenosine should be administered into a venous vessel as close as possible to the heart (vein in the antecubital bend). Regardless of the choice of procedure (pharmacologic or electrical cardioversion), the patient must be fully monitored, and the team must be ready to start advanced life-saving procedures. The literature describes two administration techniques. Standard guidelines recommended adenosine to be administered IV with an immediate bolus of normal saline solution (NSS; double-syringe technique [DST]). Only one syringe is used as part of the second technique (single-syringe technique [SST]). In this technique, adenosine is diluted with NSS. SST may be as safe as DST, equally effective for SVT termination, or even potentially more effective with the first dose. SST constitutes a simpler and more rapid approach, obviating the need for syringe-switching or using a three-way stopcock, which was actually used to administer the drug in the described case, resulting in a reduction in the margin of error in adenosine administration [14,15]. In the described case, DST was used. The literature formulates no recommendations regarding the selection of an adenosine delivery technique when intraosseous access is used by the rescuers.

The IO technique has presented itself as a possible route of adenosine administration. It is fast, reliable, requires minimal training, and can be implemented rapidly. This technique is recommended by AHA guidelines [7]. ERC indicates intravenous administration as the preferred method of adenosine administration (there is insufficient evidence supporting effective and safe IO adenosine administration) [8]. The ERC indicates exactly when intraosseous access should be performed as part of Paediatric Advanced Life Support (PALS). An attempt to establish intraosseous access should be made when the chances of obtaining peripheral access are low and already at an early stage of treatment (alternative access) [7]. Multiple studies have shown the importance and reliability of IO access in the paediatric population [16]. IO access refers to the placement a specialized hollow-bore needle through the cortex of a bone into the medullary space for infusion or for collecting laboratory samples (before submitting the sample for laboratory testing, it must be properly marked). The distal femur, proximal tibia, and distal tibia are recommended sites for IO placement in neonates and children. The preferred access site in infants and children is the anteromedial surface of the tibia, approximately 1 to 2 cm below the tibia tuberosity [17]. The choice of where to perform intraosseous access depends mainly on the device used. It seems reasonable to assume that access, as in the case of venous access, should be performed as close to the heart as possible (humeral head) due to the short half-life of adenosine in the bloodstream. However, obtaining such access will depend on the device available. Currently, pre-hospital emergency services in Poland (emergency medical services, hospital emergency departments, and HEMS) have access to both mechanical and manual devices for IO access. The mechanical devices include NIO-P (New Intraosseous—Pediatric, PerSys Medical, Houston, Tx), BIG-P (Bone Injection Gun—Pediatric, PerSys Medical, Houston, Tx) and Arrow EZ-IO (Teleflex Medical Triangle Park, NC—used by HEMS teams in Poland). Manual devices include the Jamshidi IO needle (Jamshidi, Baxter HealthCare Corporation, Deerfield, Ill) and Cook IO needle (Cook Medical, O’Halloran Road National Technology Park Limerick, Ireland). The mechanical devices used to perform IO access are more reliable, take a shorter amount of time to obtain vascular access, and require less training before they can be used correctly. An important element when selecting a device for intraosseous access is the chance of success in performing this type of access upon the first attempt. Mechanical devices are much more advantageous in this respect [18]. Regardless of whether intraosseous access is performed using a mechanical or manual device, a three-way tap will be an integral part of the access. Its use requires a certain amount of fluid inside, which may affect the therapeutic effect of adenosine, especially when SST is used (too low a drug dose). Performing intraosseous access involves certain risks. The most frequently mentioned adverse events are as follows: failure to enter the bone marrow, with extravasation or subperiosteal infusion; through-and-through penetration of the bone; osteomyelitis (rare in short-term use); physeal plate injury; local infection; skin necrosis; pain; compartment syndrome; and fat and bone microemboli (reported but are rare) [19].

One of the elements of the intraosseous access procedure is proper analgesia. When using the EZ-IO device, the manufacturer recommends 0.5 mL/kg body weight of 2% lidocaine without adrenaline and preservatives in a slow bolus (120 s), followed by 2–5 mL NS push (60 s) [20]. Although lidocaine has antiarrhythmic effects, it is not a pharmacological agent recommended by the AHA and ERC for interrupting an episode of SVT. However, its influence on the condition of the presented patient cannot be ruled out.

In the literature on intraosseous adenosine administration, there is evidence both confirming and rejecting the effectiveness of adenosine administration via this route. In 1994, research was carried out on piglets to demonstrate whether adenosine used IO was effective or not. Conclusions from this research indicated that the intraosseous route is an effective way of administering adenosine. Moreover, the peripheral venous dose required to achieve atrioventricular block is higher than the central venous dose, while the intraosseous dose falls between the central venous and peripheral venous doses of adenosine [21]. The effectiveness of adenosine in terminating supraventricular tachycardia in a paediatric patient was confirmed in the case of an 11-day-old boy described by Friedman (conversion to NSR with 100 mcg of adenosine, an early dose was given prior to attempting cardioversion, a saline flush was not reported) [22]. In 2012, a paper by Goodman and Lu was published, presenting cases of a 2-month-old and a 4-month-old child diagnosed with SVT. In both cases, the administration of adenosine did not achieve the intended result. In the first case, rhythm conversion occurred through the administration of adenosine via intravenous access (250 mcg IO, 100 mcg IV). In the second case, central access was used to administer the drug (200 mcg IO), but with SVT recurrence [5]. The cases described by Friedman and Goodman and Lu were compiled by Clark. Clark stated that intraosseous adenosine does not appear to be as effective as intravenous administration. However, he emphasizes that the reason for the failure of intraosseous administration may be the delivery technique (drug dose, NS bolus rate) [23]. In the following years, papers were published proving the effectiveness of intraosseous adenosine administration. Helleman et al. described the case of a 2-week-old boy who experienced an episode of SVT. Conversion to NSR was achieved with 0.5 mg adenosine, diluted to 3 mL NS (SST technique) [24]. Fidancı et al. reported the case of a 28-day-old boy in whom the conversion of SVT to NSR was achieved with adenosine (3450 g–0.1 mcg of adenosine per kg, single dose) [25]. The described case concerns a child older than 12 months, which makes it unique in the context of the previously described cases.

## 4. Conclusions

The described case of a paediatric patient with an episode of SVT concerns a hemodynamically unstable patient. Stable patients should be transported to the nearest paediatric cardiology department, even if tachycardia is present (depending as well on local conditions and the scope of practice and experience of the emergency team).

In the presented case report of a 3.5-year-old child, SVT was converted to NSR using two doses of adenosine, administered in accordance with ERC guidelines using the IO route. IO access was established due to difficulties in the cannulation of peripheral vessels and the technical conditions associated with transporting injured persons using a helicopter. The patient was transferred to the emergency room in good general condition.

There is no evidence in the literature that unanimously supports or rejects intraosseous adenosine administration. Further definitive studies are needed to determine whether IO administration of adenosine in stable SVT is safe and effective for paediatric patients. In future studies, special attention should be paid to the technique of intraosseous adenosine administration. It is necessary to determine whether SST or DST is more effective for adenosine administration. It is necessary to determine the recommended site for intraosseous access and the potential impact of lidocaine on the termination of an SVT episode.

## Figures and Tables

**Figure 1 healthcare-12-01509-f001:**
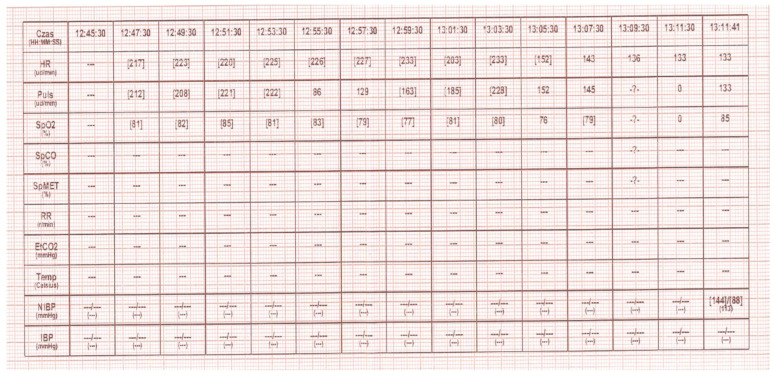
First recording of heart rate and reaction to the first dose of adenosine (Defiguard Touch 7, Schiller Schweiz AG, Bachstrasse 30, 8912 Obfelden, Switzerland). Asterisks means that no blood pressure measurement was taken during the given interval.

**Figure 2 healthcare-12-01509-f002:**
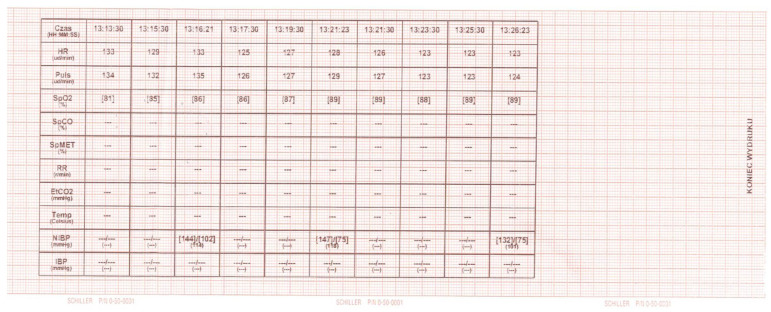
Second recording of heart rate and reaction to the second dose of adenosine (Defiguard Touch 7, Schiller Schweiz AG, Bachstrasse 30, 8912 Obfelden, Switzerland). Asterisks means that no blood pressure measurement was taken during the given interval.

## Data Availability

Data are contained within the article.

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
