# Peer review of "Successful Intraosseous (IO) Adenosine Administration for the Termination of Supraventricular Tachycardia (SVT) in a 3.5-Year-Old Child—Case Report and Literature Review"

_healthcare, 2024, doi:10.3390/healthcare12151509_

Round 1

Reviewer 1 Report

Comments and Suggestions for Authors

There are not many evidence on administration of adenosine by intraosseus route. This case report add some important information to this topic. The paper needs some substantial improvement in order to be accepted for publication:

-Please provide reference for epidemiology of SVT in pediatric group.

-The explanation of mechanism of SVT is not clear. In fact, two most frequent types of SVT are those with AV node playing an important role in mechanism of arrhythmia. 

-Atrioventricular reciprocating tachycardia is AVRT, AVNRT is Atrioventricular Node Re-Entrant Tachycardia;

-stable/unstable SVT - probably it would be better to use wording: patients in stable clinical condition with SVT/ patients in unstable clinical condition with SVT

-please provide and explanation why there is no ECG recording in your work, it is crucial to recognize the type of arrhythmia?

-could you comment on this question - Adenosin half life Is 10 sec, how could you explain its efficiency in your patient after 2 minutes?

-Case report description should focus on the topic of the paper.

-Discussion should focus on the topic of the paper.

Comments on the Quality of English Language

-Lines 45-47, please rewrite the sentence: Signs and symptoms of SVT in infants may be nonspecific such as poor feeding, vomiting, diaphoresis, hypersomia and irritability

-proper name of the device is "EZ-IO"

Author Response

REVIEWER 1.

Dear Sir/Madam,

Thank you very much for your time and valuable comments. We are convinced that the changes introduced thanks to them had a very positive impact on the quality and clarity of the article.

All major changes introduced in the manuscript are the result of an in-depth analysis of the text by the authors, influenced by the reviewers' comments. The changes aimed at improving the quality, reliability and transparency of the article concern primarily:

  1. Following the Reviewer's thoughtful suggestion, a reference was added relating to the epidemiology of SVT in the pediatric population.
  2. Referring to the Reviewer's comment, a description of the pathophysiology of SVT was added. The authors' intention was to make the article more accessible to readers, avoiding content that may be difficult for readers who do not have extensive cardiological knowledge (lines 41 – 47).
  3. The text incorrectly uses the abbreviation AVNRT to refer to Atrioventricular reciprocating tachycardia. Thank you kindly for pointing out the error.
  4. Following the reviewer's apt suggestion, the term stable/unstable was changed to patients in stable clinical condition with SVT/patients in unstable clinical condition with SVT
  5. In response to the Reviewer's question regarding the lack of an ECG record constituting the basis for the assessment of arrhythmias, reference should be made to the events on site prior to the arrival of the HEMS. In accordance with the guidelines of recognized global organizations, a twelve-lead ECG was performed. This was done on the emergency medical team's heart monitor. The record in paper form was then forwarded together with the patient to the hospital emergency department, which then to the cardiology department. The equipment available to ground teams does not have the ability to recreate events from memory. This information has been added to the body of the manuscript. Myślę, że trzeba umieścić tę informacje w manuskrypcie
  6. The authors agree with the reviewer's comment regarding the half-life of adenosine. We are extremely embarrassed because during the translation of the manuscript into English, there was a change in the wording of the sentence, which led to misleading the reader regarding the half-life of adenosine. The original wording of the text concerned the intervals in which the patient's condition was monitored using a cardiomonitor. The authors' goal was to demonstrate that rhythm conversion occurred as a result of adenosine administration, which was then documented on the attached vital sign recording. The sentence has been rephrased.
  7. According to the Reviewer's suggestion, the text has been adapted to the principles of creating a case study. We hope that the introduced changes will be satisfactory for the reviewer.
  8. Thank you for your suggestion to rephrase the sentence on lines 45-47. It has been included in the text.
  9. Thank you for pointing out the incorrect use of the EZ-IO device name. This has been changed in the manuscript.
  10.  

Thank you very much for all the reviewer's comments. We are very pleased that all comments indicated in the review were accurate and substantive. We agree with all the reviewer's comments. Authors of the manuscript have experience in the wide range of acute conditions covered by emergency medicine but do not compare to such extensive knowledge in the field of cardiology as the Reviewer. The quality of the review will significantly affect the level of the revised manuscript. We are very grateful for this.

To facilitate all Reviewers, all changes made to the manuscript are marked in yellow.

We hope that all modifications will have a positive impact on the article and the way it is perceived.

We hope that the transparent description, presentation and all changes introduced in the manuscript have alleviated the reviewer's concerns and increased the quality and reliability of the article.

Once again, thank you very much for all your comments, and suggestions. We believe that thanks to them we managed to significantly increase the quality and transparency of the article.

Reviewer 2 Report

Comments and Suggestions for Authors

please see enclosure

Comments on the Quality of English Language

The manuscript should be checked for spelling and grammar.

Author Response

Dear Sir/Madam,

Thank you very much for your time and valuable comments. We are convinced that the changes introduced thanks to them had a very positive impact on the quality and clarity of the article.

All major changes introduced in the manuscript are the result of an in-depth analysis of the text by the authors, influenced by the reviewers' comments. The changes aimed at improving the quality, reliability and transparency of the article concern primarily:

  1. Thank you for pointing out the inconsistency in the use of abbreviations and the full name in the manuscript title. The title has been corrected according to the reviewer's suggestion.
  2. A change has been made to line 37 as suggested by the reviewer. Thank you for correctly pointing out the inaccuracies in the text.
  3. The text incorrectly uses the abbreviation AVNRT to refer to Atrioventricular reciprocating tachycardia. Thank you for pointing out the error.
  4. Thank you kindly for pointing out the types of adenosine-sensitive tachycardia that the authors did not indicate. The comment has been included in the text.
  5. The child's guardian was unable to determine what diseases the child was suffering from The interview was also unreliable due to the caregivers' condition after drinking alcohol. The incident took place at a family party. The HEMS team, after an initial examination, performing a 12-lead ECG and consulting the pediatric hospital, decided on the procedure described in the manuscript. The comment has been included in the text.
  6. Thank you for pointing out the lack of description of the adenosine administration technique in the case report. The gaps have been filled.
  7. Thank you for your pertinent comments regarding the adverse effects of adenosine. The text has been corrected according to the reviewer's suggestion.

In response to the Reviewer's question regarding the lack of an ECG record constituting the basis for the assessment of arrhythmias, reference should be made to the events on site prior to the arrival of the HEMS. In accordance with the guidelines of recognized global organizations, a twelve-lead ECG was performed. This was done on the emergency medical team's heart monitor. The record in paper form was then forwarded together with the patient to the hospital emergency department, which then to the cardiology department. The equipment available to ground teams does not have the ability to recreate events from memory. The authors are unable to reconstruct from any type of arrhythmia the syndrome at the scene of the event that occurred. It should be noted that ECG monitoring was carried out at all stages of patient management. The comment has been included in the text.

Thank you very much for all the Reviewer's comments. We are very pleased that all comments indicated in the review were accurate and substantive. We agree with all the Reviewer's comments. Authors of the manuscript have experience in the wide range of acute conditions covered by emergency medicine but do not compare to such extensive knowledge in the field of cardiology as the Reviewer. The quality of the review will significantly affect the level of the revised manuscript. We are very grateful for this. When it comes to the saturation record. The authors are unable to determine what the SpO2 values indicated by the Reviewer are related to (measurement error, still limited peripheral perfusion?). The authors are unable to answer the question of why the ground team decided to wait for the HEMS team. This is part of the training of medical teams and dispatchers in order to reduce the occurrence of this type of incidents.

To facilitate all Reviewers, all changes made to the manuscript are marked in turquoise.

We hope that all modifications will have a positive impact on the article and the way it is perceived.

We hope that the transparent description, presentation and all changes introduced in the manuscript have alleviated the reviewer's concerns and increased the quality and reliability of the article.

Once again, thank you very much for all your comments, and suggestions. We believe that thanks to them we managed to significantly increase the quality and transparency of the article.

Round 2

Reviewer 1 Report

Comments and Suggestions for Authors

No comments

Comments on the Quality of English Language

Please correct minor mistake - repetition: "but there are there are" (line 54)

Author Response

Dear Sir/Madam,

Thank you once again for your time and valuable comments. The changes introduced thanks to them had a very positive impact on the quality and clarity of the article.

The indicated repetitions in lines 53-54 have been corrected at the reviewer's discretion.

To facilitate all Reviewers, all changes made to the manuscript are marked in yellow.

Once again, thank you very much for all your comments, and suggestions. We believe that thanks to them we managed to significantly increase the quality and transparency of the article.

Reviewer 2 Report

Comments and Suggestions for Authors

please see the enclosed file

Comments on the Quality of English Language

moderate editing is indicated (see as well the enclosed comments)

Author Response

Dear Sir/Madam,

Thank you very much for your time and valuable comments. We are convinced that the changes introduced thanks to them had a very positive impact on the quality and clarity of the article.

In accordance with the comments sent, we have made the following corrections:

  1. Line 121: missing information regarding the diagnostic role of adenosine has been completed.
  2. Lines 128-132 have been corrected at the reviewer's discretion.
  3. Lines 57 and 206 have been improved. The error resulted from incorrect translation.

The authors fully address the reviewer's comments regarding the need to monitor the patient during adenosine administration. This is an element covered by the guidelines of many global organizations and also taught during classes and various types of courses, and should not be missing in the proposed text. Much attention was devoted to the advantages of intraosseous access, but no possible side effects were noted. We are concerned with the possibility of transporting a stable patient with an SVT episode to the nearest center with cardiology facilities. Unfortunately, this is still a problem in the Polish health care system, which is constantly being improved through work with dispatchers and ground teams of the emergency medical services.

To facilitate all Reviewers, all changes made to the manuscript are marked in turquoise.

Once again, thank you very much for all your comments, and suggestions. We believe that thanks to them we managed to significantly increase the quality and transparency of the article.
